# Neuronal calmodulin levels are controlled by CAMTA transcription factors

**Thanh Thi Vuong-Brender[1,2†], Sean Flynn[1†], Yvonne Vallis[2], Saliha E Sönmez[2], Mario de Bono[1]\***

[1]Cell Biology Division, Medical Research Council Laboratory of Molecular Biology, Cambridge, United Kingdom; [2]Institute of Science and Technology Austria (IST Austria), Klosterneuburg, Austria

**Abstract** The ubiquitous $Ca^{2+}$ sensor calmodulin (CaM) binds and regulates many proteins, including ion channels, CaM kinases, and calcineurin, according to $Ca^{2+}$-CaM levels. What regulates neuronal CaM levels, is, however, unclear. CaM-binding transcription activators (CAMTAs) are ancient proteins expressed broadly in nervous systems and whose loss confers pleiotropic behavioral defects in flies, mice, and humans. Using *Caenorhabditis elegans* and *Drosophila*, we show that CAMTAs control neuronal CaM levels. The behavioral and neuronal $Ca^{2+}$ signaling defects in mutants lacking *camt-1*, the sole *C. elegans* CAMTA, can be rescued by supplementing neuronal CaM. CAMT-1 binds multiple sites in the CaM promoter and deleting these sites phenocopies *camt-1*. Our data suggest CAMTAs mediate a conserved and general mechanism that controls neuronal CaM levels, thereby regulating $Ca^{2+}$ signaling, physiology, and behavior.

**\*For correspondence:**
mdebono@ist.ac.at

[†]These authors contributed equally to this work

**Competing interests:** The authors declare that no competing interests exist.

## Introduction

Calmodulin-binding transcription activators (CAMTAs) are a highly conserved family of CaM-binding transcription activators (*Finkler et al., 2007*). In plants, CAMTAs mediate transcriptional changes in response to $Ca^{2+}$ signals evoked by biotic and abiotic stress (*Yang and Poovaiah, 2002*; *Du et al., 2009*; *Doherty et al., 2009*; *Pandey et al., 2013*; *Shkolnik et al., 2019*). Mammals encode two CAMTA proteins, CAMTA1 and CAMTA2, respectively enriched in the brain and heart (*Song et al., 2006*). Loss of CAMTA1 in the mouse nervous system leads to defects in hippocampal-dependent memory formation, degeneration of cerebellar Purkinje cells and ataxia (*Long et al., 2014*; *Bas-Orth et al., 2016*). Humans heterozygous for lesions in the *CAMTA1* gene exhibit a range of neurological phenotypes, including intellectual disability, cerebellar ataxia, and reduced memory performance (*Huentelman et al., 2007*; *Thevenon et al., 2012*; *Shinawi et al., 2015*). Mechanistically, however, little is known about the origin of these neuro-behavioral phenotypes.

CaM is a ubiquitously expressed $Ca^{2+}$ binding protein that plays a key role in transducing responses to $Ca^{2+}$ changes (*Faas et al., 2011*; *Baimbridge et al., 1992*). $Ca^{2+}$-CaM modifies a host of neuronal functions, including signal transduction, ion currents, vesicle fusion, learning and memory, metabolism, and apoptosis (*Hoeflich and Ikura, 2002*; *Berchtold and Villalobo, 2014*), by regulating dozens of binding targets including the CaM kinases, calcineurin, and diverse ion channels (*Wayman et al., 2008*; *Saimi and Kung, 2002*). CaM levels are thought to be limiting compared to the combined concentration of $Ca^{2+}$-CaM binding proteins (*Sanabria et al., 2008*), and relatively small changes in CaM levels are predicted to impact $Ca^{2+}$-CaM regulation of downstream targets (*Pepke et al., 2010*). What mechanisms regulate neuronal CaM levels is, however, unclear. We identify CAMTA as a key regulator of CaM expression in multiple neuron types, and in both *Caenorhabditis elegans* and *Drosophila*, and suggest that it is a general and conserved regulator of $Ca^{2+}$/CaM signaling in nervous systems.

## Results

### CAMT-1 functions in neurons to regulate multiple behaviors

Most natural isolates of *C. elegans* feed in groups. By contrast, the standard *C. elegans* lab strain, N2, feeds alone, due to a gain-of-function mutation in a neuropeptide receptor called NPR-1 (*de Bono and Bargmann, 1998*). Using *npr-1(ad609)* null mutants of the N2 strain (denoted as *npr-1* throughout this manuscript), which aggregate strongly (*Figure 1—figure supplement 1A*), we performed a forward genetic screen for genes required for aggregation (*Chen et al., 2017*). The screen identified multiple aggregation-defective strains with mutations in *camt-1*, the sole *C. elegans* CAMTA (*Figure 1—figure supplement 1A*).

Aggregation is closely linked to escape from normoxia (21% $O_2$) (*Busch et al., 2012*; *Rogers et al., 2006*; *Gray et al., 2005*). Normoxia elicits rapid movement in *npr-1* animals, which is rapidly suppressed when $O_2$ levels drop (*Figure 1A*). Since aggregating animals create a local low $O_2$ environment, due to aerobic respiration, an animal encountering a group from normoxia switches from fast to slow movement, thereby staying in the group. *camt-1* mutants showed defective responses to $O_2$ stimuli. Compared to *npr-1* controls, animals from a mutant strain isolated in the screen, *camt-1(db973); npr-1*, which harbors a premature stop codon in CAMT-1 (Q222\*), were hyperactive in 7% $O_2$, and showed reduced arousal when switched from 7% to 21% $O_2$ (*Figure 1A–B*). A deletion (*The C. elegans Deletion Mutant Consortium, 2012*) that removed 451 residues of CAMT-1, *camt-1(ok515)*, conferred similar defects (*Figure 1A–B*). A fosmid transgene containing a wild-type (WT) copy of the *camt-1* genomic locus rescued *camt-1* mutant phenotypes, restoring fast movement at 21% $O_2$, and slow movement at 7% $O_2$ (*Figure 1C*). These results indicate that CAMT-1 is required for *C. elegans* to respond appropriately to different $O_2$ levels.

CAMT-1 has the characteristic domain architecture of CAMTAs (*Finkler et al., 2007*): a DNA-binding domain (CG-1), an immunoglobulin-like fold (IPT/TIG) similar to those found in non-specific DNA-binding/dimerization domains of other transcription factors, ankyrin repeats (ANKs), a putative $Ca^{2+}$-dependent CaM-binding domain (CaMBD) and multiple IQ motifs that are thought to bind CaM in a $Ca^{2+}$-independent manner (*Figure 1B*, *Figure 1—figure supplement 1B–C*; *Bouché et al., 2002*; *Choi et al., 2005*). CAMT-1 also has predicted nuclear localization and nuclear export signals (NLS/NES, *Figure 1B*).

In mice, humans, and flies, CAMTA transcription factors are expressed in many brain regions (*Huentelman et al., 2007*; *Bas-Orth et al., 2016*; *Sato et al., 2019*; *Long et al., 2014*). We generated a fosmid-based reporter to map the expression pattern of the longest isoform of *C. elegans* CAMTA, CAMT-1a. This fluorescent reporter was functional, as it rescued the behavioral defects of *camt-1* mutants (*Figure 1C*), and revealed that CAMT-1 was expressed broadly and specifically in the nervous system (*Figure 1D*). We observed CAMT-1 expression in sensory neurons with exposed ciliated endings, motor neurons of the ventral cord, the URX $O_2$-sensing neuron, and URX's post-synaptic partner, the RMG hub interneurons (*Figure 1—figure supplement 2*). *camt-1*'s broad expression prompted us to ask if *camt-1* mutants display pleiotropic behavioral phenotypes. We asked whether CAMT-1 is required for other aversive behaviors, such as avoidance of $CO_2$, or for chemoattraction to odors and salts. In response to a rise in $CO_2$, WT control (N2) worms transiently perform omega turns, $\Omega$-shaped body bends that re-orient the animal away from the stimulus (*Bretscher et al., 2008*). *camt-1* mutants exhibited abnormally high levels of $\Omega$-turns without a $CO_2$ stimulus and a prolonged increase in $\Omega$-turns in response to a rise in $CO_2$ (*Figure 1E*). *C. elegans* avoids $CO_2$ but is attracted toward salt and a range of volatile compounds (*Ward, 1973*; *Bargmann et al., 1993*). Chemotaxis toward NaCl and odorant attractants such as benzaldehyde and diacetyl was reduced in *camt-1* mutants, and these defects were rescued by a fosmid transgene containing WT CAMT-1 (*Figure 1F*). Taken together, these data show that CAMT-1 function is important for multiple *C. elegans* behaviors.

Many deleterious human alleles of CAMTA1 alter the CG-1 DNA-binding domain (*Thevenon et al., 2012*). To assess the importance of the putative DNA-binding domain of CAMT-1, we used CRISPR-Cas9 to engineer mutations in conserved residues of the CG-1 domain (*Figure 1—figure supplement 1B*). These mutants showed defects in aggregation and in their response to $O_2$, recapitulating phenotypes of the *camt-1* deletion mutants described above (*Figure 1G*, *Figure 1—figure supplement 1B*). These results suggest that CAMT-1 binding to DNA is essential for its function, at least for $O_2$ escape behavior.

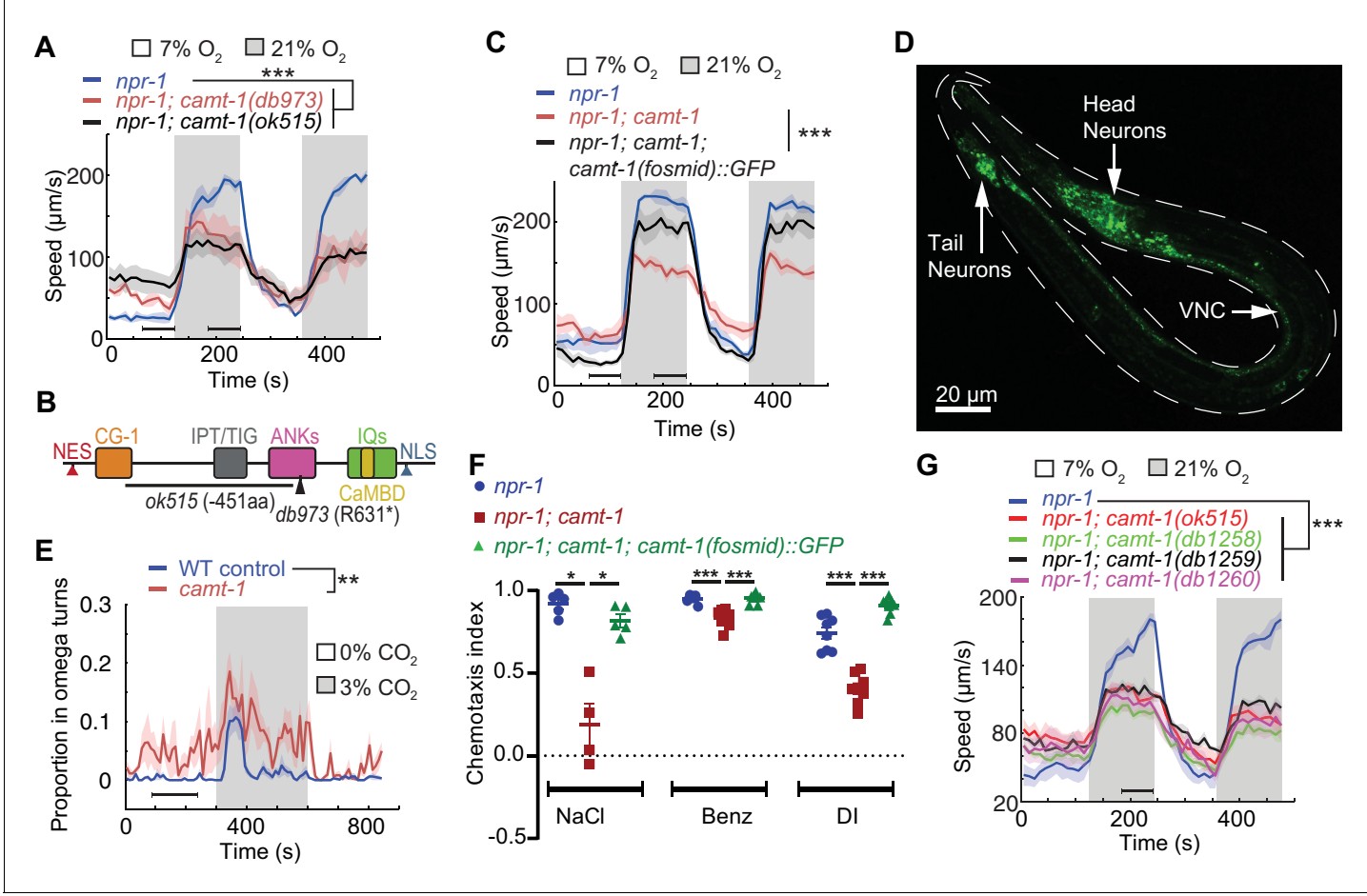

**Figure 1.** *camt-1* mutants exhibit pleiotropic behavioral defects. (A) *camt-1(db973)* and *camt-1(ok515)* mutants (see also (B)) exhibit altered locomotory responses to 21% $O_2$ and hyperactive movement at 7% $O_2$. (B) The domain organization of CAMT-1, highlighting *camt-1* loss of function mutations used in this study. (C) A WT copy of the *camt-1* genomic locus rescues the $O_2$-response defects of *camt-1(db973)* mutants. (D) CAMT-1a::GFP driven from its endogenous regulatory sequences in a recombineered fosmid is expressed widely in the nervous system. (E) *camt-1(db973)* mutants exhibit an increased turning frequency both in the presence and absence of a $CO_2$ stimulus. Assays were performed in 7% $O_2$. (F) *camt-1(ok515)* mutants show defects in chemotaxis to NaCl, benzaldehyde (Benz), and diacetyl (DI), which can be rescued by expressing a WT copy of CAMT-1. Colored bars indicate the mean and error bars indicate the SEM. (G) The $O_2$-response defects of mutants harboring amino acid substitutions in the CG-1 DNA-binding domain (*db1258, db1259*, and *db1260* alleles; see also *Figure 1—figure supplement 1B*), are comparable to those of a *camt-1(ok515)* deletion mutant. (B, C, E, G) Lines indicate average speed and shaded regions SEM, black horizontal bars indicate time points used for statistical tests. (B, C, E–G) Mann-Whitney U-test, ns: p≥0.05, *: p<0.05, **: p<0.01, ***: p<0.001. Number of animals: n≥22 (A), n>41 (C), n≥23 (E), n≥4 assays for each genotype (F), n≥56 (G). ANK, ankyrin domain; CaMBD, calmodulin-binding domain; CG-1, DNA-binding domain; IPT/TIG, Ig-like, plexins, transcription factors or transcription factor immunoglobulin; IQ, calmodulin-binding motif; NES, nuclear export signal; NLS, nuclear localization signal; VNC, ventral nerve cord; WT, wild-type.

The online version of this article includes the following figure supplement(s) for figure 1:

**Figure supplement 1.** CAMT-1 structure.

**Figure supplement 2.** CAMT-1 is widely expressed in the nervous system.

We targeted CAMT-1 cDNA expression to different subsets of neurons in the neuronal circuit regulating the response to $O_2$, to find out where CAMT-1 is required to promote aerotaxis. $O_2$ is sensed mainly by the sensory neurons URX, AQR, and PQR, and tonic signaling from URX to RMG drives high locomotory activity at 21% $O_2$ (*Busch et al., 2012*; *Zimmer et al., 2009*). Selectively expressing CAMT-1 to the RMG hub interneurons, but not $O_2$ sensing neurons, rescued the fast movement at 21% $O_2$ of *camt-1* mutants (*Figure 1—figure supplement 1D–E*). The defective response of *camt-1* mutants to 7% $O_2$ was not rescued by expressing CAMT-1 in RMG, or by simultaneous expression in RMG and $O_2$-sensing neurons (*Figure 1—figure supplement 1D–E*). These data are consistent with

a model in which CAMT-1 acts in multiple neurons. As expected, pan-neuronal expression rescued *camt-1* mutant phenotypes, and expression of the isoform a alone (CAMT-1a) was sufficient for rescue (*Figure 2A*).

CAMTA transcription factors bind and can be regulated by CaM (*Yang and Poovaiah, 2002*; *Du et al., 2009*; *Doherty et al., 2009*; *Pandey et al., 2013*; *Shkolnik et al., 2019*). $Ca^{2+}$-CaM dependent changes in gene expression are known to be important for both the development and function of the nervous system (*West et al., 2002*; *Chin and Means, 2000*). To test whether CAMT-1 activity is essential during development, we expressed CAMT-1 cDNA from a heat-shock-inducible promoter. Without heat-shock, this transgene did not rescue the hyperactivity phenotype of *camt-1* mutants (*Figure 2B*). By contrast, inducing CAMT-1 expression in the last larval stage/young adults rescued the aggregation (data not shown) and speed response defects, albeit not completely (*Figure 2C*), suggesting that CAMT-1 can function in adults post-developmentally to regulate behavioral responses to ambient $O_2$.

## CAMT-1 dampens $Ca^{2+}$ responses in sensory neurons

To test whether disrupting *camt-1* altered physiological responses to sensory cues we used Yellow Cameleon (YC) $Ca^{2+}$ sensors to record stimulus-evoked $Ca^{2+}$ changes in the URX $O_2$-sensor, and in the BAG and AFD neurons, which respond to $CO_2$. BAG drives omega turns when $CO_2$ levels rise (*Bretscher et al., 2011*; *Hallem and Sternberg, 2008*). Expressing YC sensors in these neurons did not alter the response of animals to $O_2$ or $CO_2$ (*Figure 3—figure supplement 1A–C*). We found that baseline $Ca^{2+}$ and stimulus-evoked $Ca^{2+}$ responses in URX, BAG, and AFD neurons were significantly elevated in *camt-1* mutants across all the $O_2/CO_2$ conditions we tested (*Figure 3A–B*, *Figure 3—figure supplement 1D*). These data suggest that CAMT-1 activity somehow dampens the $Ca^{2+}$ responses of these sensory neurons. We obtained similar results for $Ca^{2+}$ measurements in BAG using a $Ca^{2+}$ reporter, TN-XL (*Bazopoulou et al., 2017*; *Mank et al., 2006*), which uses chicken troponin C instead of CaM to bind $Ca^{2+}$ (*Figure 3—figure supplement 1E–F*). We observed the converse phenotype, reduced $Ca^{2+}$ baselines and responses, when we overexpressed CAMT-1 cDNA specifically in $O_2$ sensors or in BAG neurons of control animals (*Figure 3C–D*). Overexpressing CAMT-1 slightly reduced expression from the *gcy-37* promoter we used to express YC in $O_2$ sensors, as measured using a *gcy-37p::gfp* reporter (*Figure 3—figure supplement 1G*). Although we cannot completely exclude that this contributes to the reduced baseline YFP/CFP ratio, we note that

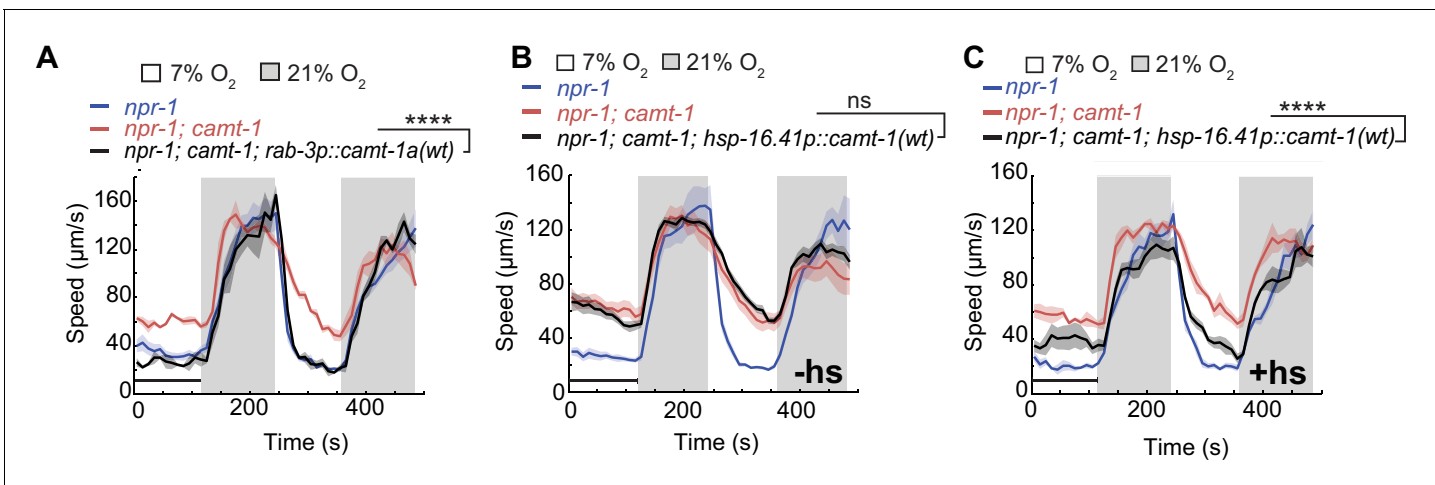

**Figure 2.** CAMT-1 acts in neurons and is not required developmentally to regulate the oxygen response. (A) Pan-neuronal expression using the *rab-3* promoter of the longest CAMT-1 isoform, CAMT-1a, in *camt-1(db973)* mutants, rescues $O_2$ response defects. (B, C) Transgenic expression of CAMT-1 from the *hsp-16.41* heat-shock promoter does not rescue the hyperactive locomotion of *camt-1(ok515)* mutants at 7% $O_2$ without heat-shock (B). Heat-shock-induced expression of CAMT-1 in L4 animals rescues this phenotype in *camt-1(ok515)* mutants, although partially (C). Lines indicate average speed and shaded regions SEM. Black horizontal bars indicate time points used for statistical tests. Mann-Whitney U-test, ns: p≥0.05, ***: p<0.001. Number of animals: n≥39 (A), n≥158 (B), n≥56 (C). hs, heat-shock.

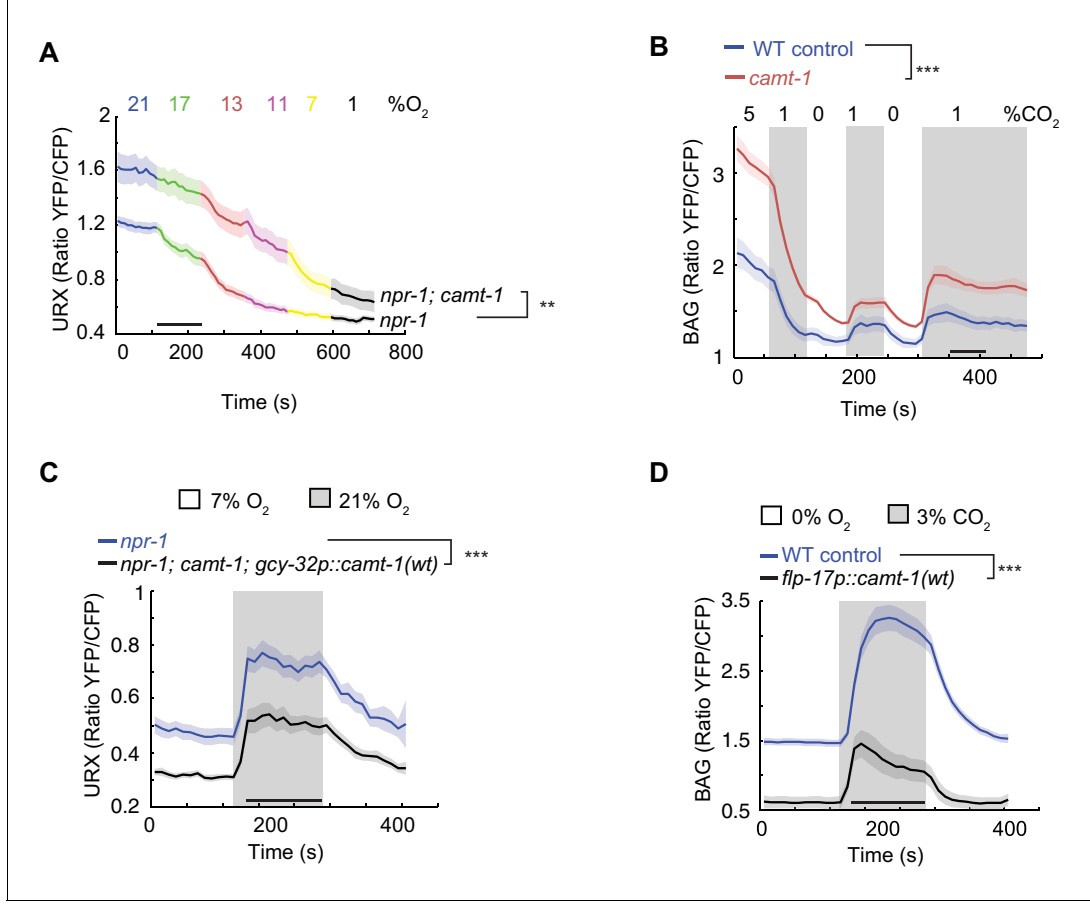

**Figure 3.** *camt-1* mutants show altered Ca²⁺ traces in sensory neurons. (**A, B**) The URX O₂-sensing neurons (**A**) and the BAG CO₂ sensors (**B**) show higher Ca²⁺ baselines and Ca²⁺ responses across a range of stimulus intensities in *camt-1(db973)* mutants. (**C–D**) Overexpressing wild-type *camt-1* cDNA in O₂-sensing (using *gcy-32p*, **C**) or BAG neurons (using *flp-17p*, **D**) strongly reduces Ca²⁺ levels in these neurons. n≥15 (**A**), n≥18 (**B**), n≥17 (**C**), and n≥20 animals (**D**). Strains express a Yellow Cameleon sensor in O₂-sensing neurons (**A, C**), or in BAG (**B, D**) (see Materials and methods). Average YFP/CFP ratios (line) and SEM (shaded regions) are plotted. **: $p<0.01$, ***: $p<0.001$, Mann-Whitney U-test.
The online version of this article includes the following figure supplement(s) for figure 3:

**Figure supplement 1.** Analyses using Ca²⁺ imaging lines.

---

cameleon is a ratiometric sensor. Taken together, our results suggest that *camt-1* regulates the excitability of sensory neurons.

## Calmodulin is one of only two genes whose expression is regulated by CAMT-1 across all neuronal types profiled

To identify downstream targets of CAMT-1, we compared the transcriptional profiles of multiple neural types in *camt-1; npr-1* and *npr-1* control animals (*Kaletsky et al., 2018*). We separately profiled the O₂-sensors URX/AQR/PQR, the RMG interneurons, the AFD thermosensors, and the BAG O₂/CO₂ sensors. We collected the neurons using FACS from strains in which they were labeled with GFP, and performed 4–10 biological replicates for robust statistical power. Analysis of the data revealed altered expression of many genes, with most changes being neural-type specific (*Figure 4A*, *Supplementary files 1* and *2*). A striking exception was *cmd-1* (*c*almo*d*ulin-1), encoding *C. elegans* CaM. *cmd-1* was one of only two genes whose expression was reduced in all four neural profiles relative to WT controls. The other gene, Y41C4A.17, has no known homolog in mammals.

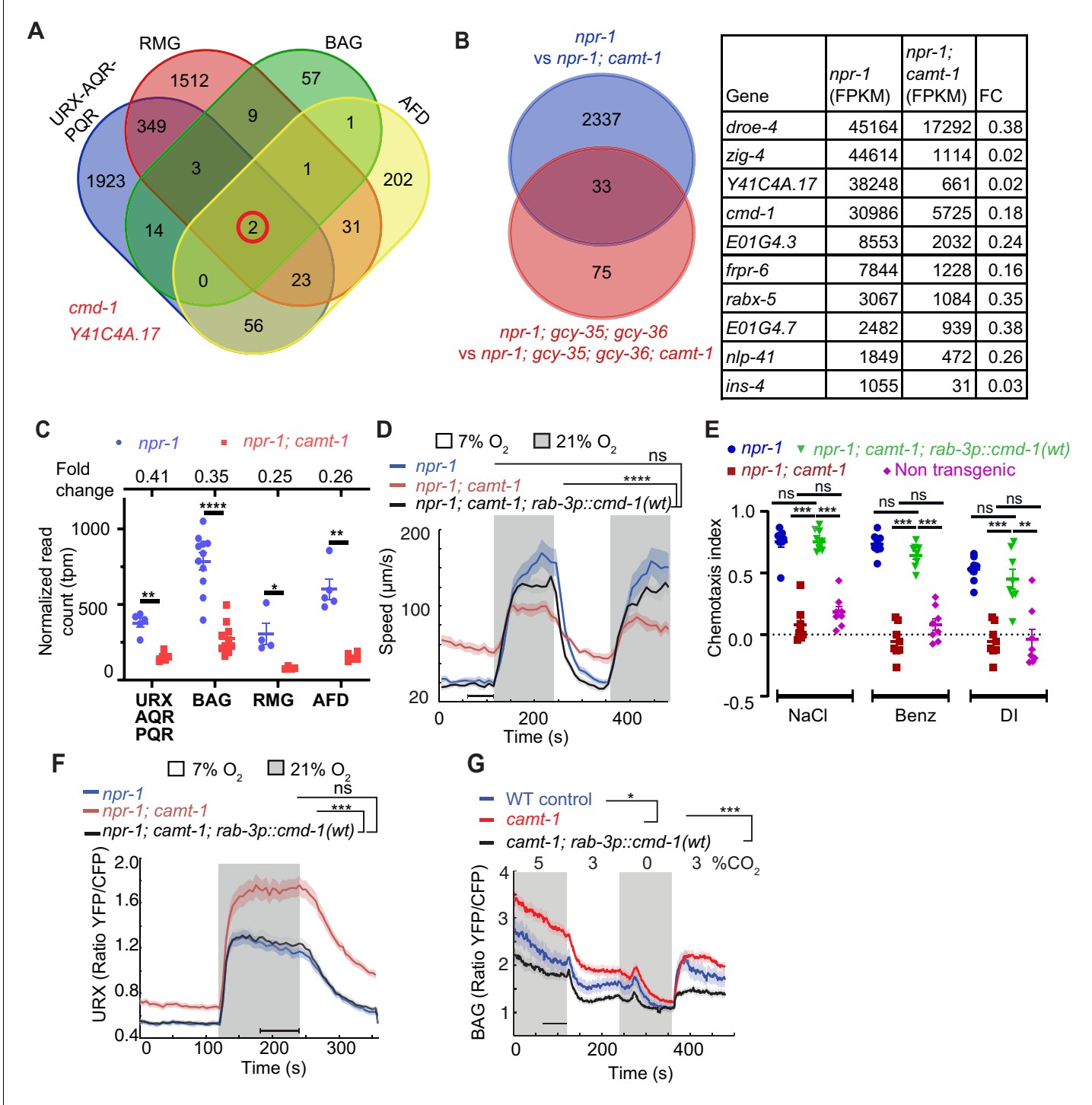

**Figure 4.** The pleiotropic phenotypes of *camt-1* reflect a role in regulating the expression of calmodulin. (A) Venn diagram showing numbers of genes differentially regulated by CAMT-1 in neuron types we profiled (URX/AQR/PQR, BAG, AFD, and RMG). Two genes, *cmd-1* (*calmodulin-1*) and *Y41C4A.17*, show consistently altered expression in all neural types profiled. (B) Left: Venn diagram comparing the number of genes differentially regulated by CAMT-1 in URX/AQR/PQR neurons in *npr-1* versus *npr-1; gcy-35; gcy-36* genetic backgrounds. Right: The most highly expressed genes (read count>1000 FPKM) among the 33 loci regulated by CAMT-1 across all genotypes tested. (C) *cmd-1* transcript read counts and FC (top) for URX/AQR/PQR, BAG, AFD, and RMG neurons in *camt-1* mutants compared to controls. Each dot or square represents a separate RNA Seq experiment. (D, E) Supplementing CMD-1 expression in neurons using a *rab-3p::cmd-1(wt)* transgene rescues the $O_2$-response (C) and chemotaxis (D) phenotypes of *camt-1* mutants. (F, G) Supplementing CMD-1 expression in neurons also rescues the *camt-1* $Ca^{2+}$-response phenotypes of URX neurons to $O_2$ (F) and of BAG neurons to $CO_2$ (G). Responses to $CO_2$ were assayed in 7% $O_2$. ns: p≥0.05, *: p<0.05, **: p<0.01, ***: p<0.001, ****: p<0.0001, Mann-
*Figure 4 continued on next page*

Figure 4 continued

Whitney U-test (C–H). n≥4 replicates for all cell types (A, B, C), n≥103 (D), n=8 assays for each condition (E), n=32 for each genotype (F), n≥58 animals (G). *camt-1* denotes *camt-1(ok515)*. (D, F, G) Lines represent average speed and shaded regions the SEM, black horizontal bars indicate time points used for statistical tests. (C, E) Colored bars indicate the mean and error bars indicate the SEM. FC: fold change.

The online version of this article includes the following figure supplement(s) for figure 4:

**Figure supplement 1.** Neuronal calmodulin levels modify behavioral responses to $O_2$.

## Most *camt-1*-dependent gene expression changes in $O_2$ sensing neurons are associated with altered neural activity

Altered $Ca^{2+}$ signaling can drive changes in neuronal gene expression (*Yap and Greenberg, 2018*). This prompted us to investigate if the altered $Ca^{2+}$ signaling we observed in *camt-1* mutants contributed to the altered gene expression. To address this, we focused on the URX/AQR/PQR $O_2$ sensors, which showed the altered expression of 2370 genes in *camt-1* mutants. Our profiling experiments were carried out in normoxia, when these neurons exhibit tonic high $Ca^{2+}$ levels due to sustained cGMP signaling mediated by a heterodimeric soluble guanylate cyclase composed of GCY-35 and GCY-36 subunits, which binds and is activated by $O_2$ (*Zimmer et al., 2009*; *Couto et al., 2013*). Disrupting GCY-35 or GCY-36 abolishes the $O_2$ response and causes these neurons to have a constitutive low baseline $Ca^{2+}$ (*Zimmer et al., 2009*). We therefore compared the number of genes differentially regulated in URX/AQR/PQR neurons that we isolated and sorted from *gcy-35; gcy-36; npr-1* and *gcy-35; gcy-36; npr-1; camt-1* mutant animals. We only observed 108 differentially regulated genes between these genotypes, a dramatic decrease from the 2370 genes we observed when we compared the same neurons between *npr-1* and *npr-1; camt-1*. Out of the 108 genes, 33 genes are common across the two sets of comparisons (*Figure 4B*). Sorting these 33 genes in decreasing order of expression (Table in *Figure 4B*), we found that they included *cmd-1* and *Y41C4A.17,* the two genes regulated by *camt-1* in all neuronal types we profiled. These results support the hypothesis that most of the genes expression changes we observe in $O_2$ sensing neurons in *camt-1* mutants are due to altered $Ca^{2+}$ signaling rather than direct control by CAMT-1, but that *cmd-1*, encoding CaM, is an exception.

## CAMT-1 phenotypes reflect reduced expression of calmodulin

CaM regulates many functions in the nervous system, including excitability (*Wayman et al., 2008*; *Zalcman et al., 2018*). The levels of CaM mRNA in *camt-1* mutants was 2.5- to 4-fold lower than in controls, depending on neural type (*Figure 4C*). We speculated that most *camt-1* phenotypes could be due to reduced CMD-1/CaM expression. Straightforward comparison of *camt-1* and *cmd-1* loss of function phenotypes was not possible, since disrupting *cmd-1* confers lethality (*Karabinos et al., 2003*; *Au et al., 2019*). We therefore, asked if supplementing CMD-1/CaM expression in *camt-1* mutants, using a pan-neuronal promoter (*rab-3p*), could rescue *camt-1* phenotypes. We made four transgenic lines that expressed CMD-1 to different levels (*Figure 4—figure supplement 1A*). To monitor expression, we placed sequences encoding mCherry in an operon with *cmd-1* (noted as *cmd-1::SL2::mCherry,* see Materials and methods). The *rab-3p::cmd-1::SL2::mCherry* transgene expressing the lowest levels of fluorescence (line A, *Figure 4—figure supplement 1A*) strongly rescued the abnormal $O_2$-escape response of *camt-1* mutants (*Figure 4D*). Further increasing CMD-1 expression levels restored quiescence behavior in animals kept at 7% $O_2$ but progressively reduced the speed attained at 21% $O_2$ (*Figure 4—figure supplement 1B*).

Supplementing CMD-1 in the nervous system using the lowest expressing *rab-3p::cmd-1::SL2:: mCherry* line also restored normal chemotaxis toward salt, benzaldehyde, and diacetyl in *camt-1* mutants (*Figure 4E*), and rescued the hyperexcitability defects in URX and BAG neurons of *camt-1* mutants (*Figure 4F–G*). By contrast, deleting the entire coding region of Y41C4A.17 did not affect aggregation of *npr-1* animals (data not shown). Our data suggest that reduced CMD-1 expression accounts for *camt-1* $Ca^{2+}$ signaling and behavioral defects (see also below).

## CAMTA promotes CaM expression in *Drosophila melanogaster*

Fly mutants of CAMTA show slow termination of photoresponses compared to WT controls (*Han et al., 2006*), and also exhibit defects in male courtship song (*Sato et al., 2019*). An allele of

the *Drosophila* CaM gene that deletes part of the promoter and reduces CaM expression also shows slow termination of photoresponses (*Scott et al., 1997*). This phenotypic similarity, and our findings in *C. elegans*, prompted us to ask if CAMTA promotes CaM expression in flies too. We obtained two characterized alleles of *Drosophila* CAMTA (*dCAMTA*), *tes*$^2$ and *cro*, which respectively contain an L1420Stop mutation and a transposon insertion (*Han et al., 2006*; *Sato et al., 2019*). *tes*$^2$ mutants showed a modest decrease in dCAMTA mRNA level, suggesting that the premature stop late in the protein does not induce mRNA degradation (*Figure 5—figure supplement 1*). The level of dCAMTA mRNA was strongly reduced in *cro* mutants as reported previously (*Sato et al., 2019*; *Figure 5—figure supplement 1*). We assessed the levels of CaM mRNA and CaM in the heads of dCamta mutant flies using quantitative RT-PCR and Western blots. Each method reported significant decreases in CaM expression compared to controls in both *tes*$^2$ and *cro* mutant flies (*Figure 5A–C*). Moreover, immunostaining dissected retinas from *cro* mutants showed reduced CaM expression in rhabdomeres (*Figure 5D–E*). These results suggest that the transcriptional upregulation of neuronal CaM by CAMTA is conserved from worms to flies.

## CAMT-1 directly regulates CMD-1/CaM transcription through multiple binding sites at the *cmd-1/CaM* promoter

To test whether CAMT-1 directly regulates *C. elegans* CaM expression by binding the *cmd-1* promoter, we performed chromatin immunoprecipitation sequencing (ChIP-seq) using a CRISPR-knock-in CAMT-1a::GFP strain. Our analysis revealed about 200 loci that were significantly enriched in CAMT-1a::GFP pulldowns compared to input, and to a mock pulldown (*Supplementary file 3*). At the top of the list was *cmd-1*: we observed three peaks at ~6.3 kb, 4.8 kb, and 2.2 kb upstream of the CMD-1 translation start site in the CAMT-1a::GFP pulldown experiments (*Figure 6A*, *Figure 6—figure supplement 1A*). We called these peaks A, B, and C, respectively. Thus, CAMT-1 is recruited to multiple sites upstream of *cmd-1*. A CAMT-1 binding peak was also found in the promoter region of Y41C4A.17, the only other gene whose expression was reduced in all the neurons profiled from *camt-1* mutants (*Figure 6—figure supplement 1B*).

To test whether the CAMT-1 ChIP-seq peaks in the *cmd-1* promoter region regulated CMD-1 transcription, we generated CRISPR strains that deleted one or more of these peaks. A strain harboring 110 bp and 136 bp deletions at peaks B and C, respectively (*Figure 6A–B*, *db1275)*, and a strain harboring a 200 bp deletion at peak A (*Figure 6A–B*, *db1280*) exhibited aggregation and O$_2$ escape responses similar to *npr-1* mutant controls (*Figure 6B*). However, a strain harboring all three deletions (*Figure 6A–B*, *db1278*) exhibited strong aggregation defects (*Figure 1—figure supplement 1A*) and defects in the locomotory responses to O$_2$ that mirrored those of *camt-1* loss-of-function mutants (*Figure 6B*, *Figure 1A*). Notably, the hyperactivity at 7% O$_2$ of *db1278* mutants could be rescued by expressing additional CMD-1 in the nervous system. Like *camt-1(ok515)* mutants, *cmd-1 (db1278)* mutants also showed chemotaxis defects toward salt, benzaldehyde, and diacetyl that could be rescued by supplementing neuronal expression of CMD-1 (compare *Figures 4D* and *6C*). These results suggest that CAMT-1 binds multiple sites in the CMD-1 promoter and acts redundantly at these sites to promote neuronal CaM expression.

## Calmodulin can inhibit its own expression via CAMT-1

CaM is a key regulator of neural function. We speculated that CMD-1/CaM might homeostatically regulate its own expression via a negative feedback loop. To investigate this hypothesis, we built a transcriptional reporter for *cmd-1* by fusing the 8.9 kb DNA fragment immediately upstream of the CMD-1 translational start site to sequences encoding GFP. This reporter showed strong fluorescence expression in neurons and muscle, including pharyngeal muscle (*Figure 7—figure supplement 1A*). We next introduced this reporter (*cmd-1p::gfp*) into a *C. elegans* line that overexpressed CMD-1/CaM in neurons, using the *rab-3* promoter (*rab-3p::cmd-1*), and measured neuronal GFP fluorescence in single (*cmd-1p::gfp*) and double (*cmd-1p::gfp+rab-3p::cmd-1*) transgenic animals. We normalized expression using pharyngeal GFP levels. Animals expressing *rab-3p::cmd-1* reduced neuronal expression of GFP from the *cmd-1p::gfp* reporter. These data suggest that the high levels of CMD-1 can repress expression from the *cmd-1* promoter (*Figure 7—figure supplement 1B*). To examine if this repression is achieved via CaM binding to CAMT-1, we introduced into the double transgenic background a *camt-1* allele that disrupts the 4 IQ domains, noted as *camt-1(4IQ\*)*

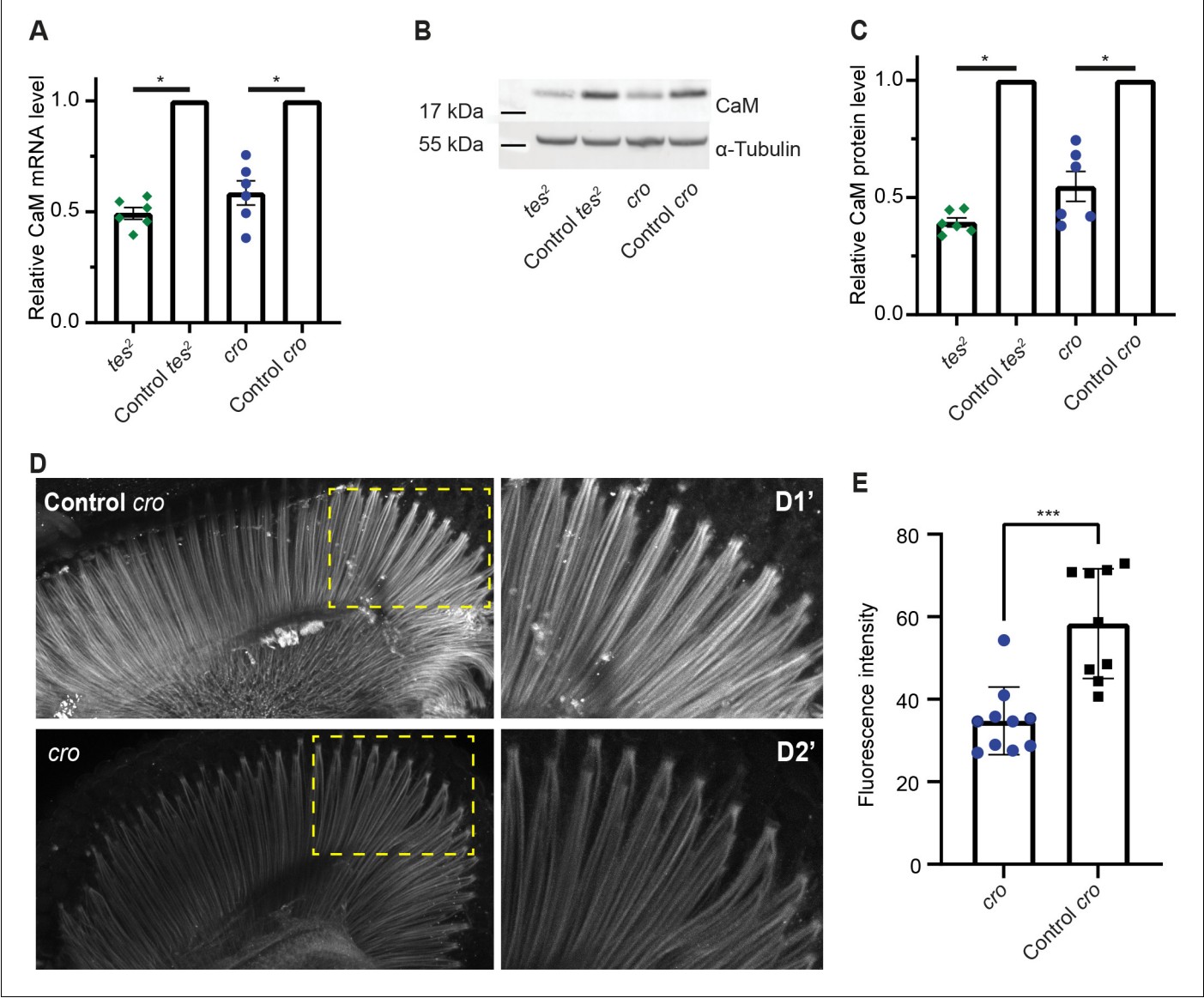

**Figure 5.** CAMTA regulates CaM expression in *Drosophila*. (A) The *Drosophila* CAMTA mutants *tes²* and *cro* show decreased CaM mRNA levels compared to control flies. mRNA levels in fly heads were measured by quantitative PCR. CAMTA mRNA levels were first normalized to *RpL32* (*rp49*), the qPCR internal control, and then to the value of control flies. (B, C) *tes²* and *cro* mutants show a decrease in CaM protein levels compared to control flies. Protein levels were determined using Western blot of proteins extracted from fly heads. (B) shows a representative picture and (C) shows quantification. CAMTA protein levels were first normalized to alpha-tubulin levels, then to the value of the control flies. (D, E) Immunostaining of fly retinae using CaM antibodies shows reduction of staining of rhabdomeres in *cro* mutants (see also *Figure 5—figure supplement 1B*). (D) shows representative pictures of control and *cro* retinae, respectively, with D1' and D2' are blow-ups of yellow rectangle in the left pictures. (E) shows quantification of CaM intensity. (A, C) *: p<0.05, one sample Wilcoxon test to control value of 1, n=6 for each genotype, colored bars indicate the mean and error bars indicate the SEM. (E) ***: p<0.001, Mann-Whitney U-test. $w^{1118}$; $cn^1$ and $w^{1118}$; *sb* are control flies for *tes²* ($w^{1118}$; $cn^1$; *tes²*) and *cro* ($w^{1118}$; *cro*; *sb*) mutants, respectively. CaM, calmodulin; CAMTA, CaM-binding transcription activator.
The online version of this article includes the following figure supplement(s) for figure 5:

**Figure supplement 1.** CAMTA and CaM expression in *Drosophila*.

(*Figure 1—figure supplement 1B*). In this allele, codons encoding the conserved isoleucine residues in the four putative IQ domains of CAMT-1 were mutated to codons that encode asparagines. The *camt-1(4IQ\*)* allele did not disrupt the $O_2$-avoidance behaviors of *npr-1* mutant animals (*Figure 7—figure supplement 1C*), suggesting that CaM binding to CAMT-1 via the IQ motifs is not essential for $O_2$ escape behavior. By contrast, we found that *camt-1(4IQ\*)* animals expressing *cmd-1p::gfp*

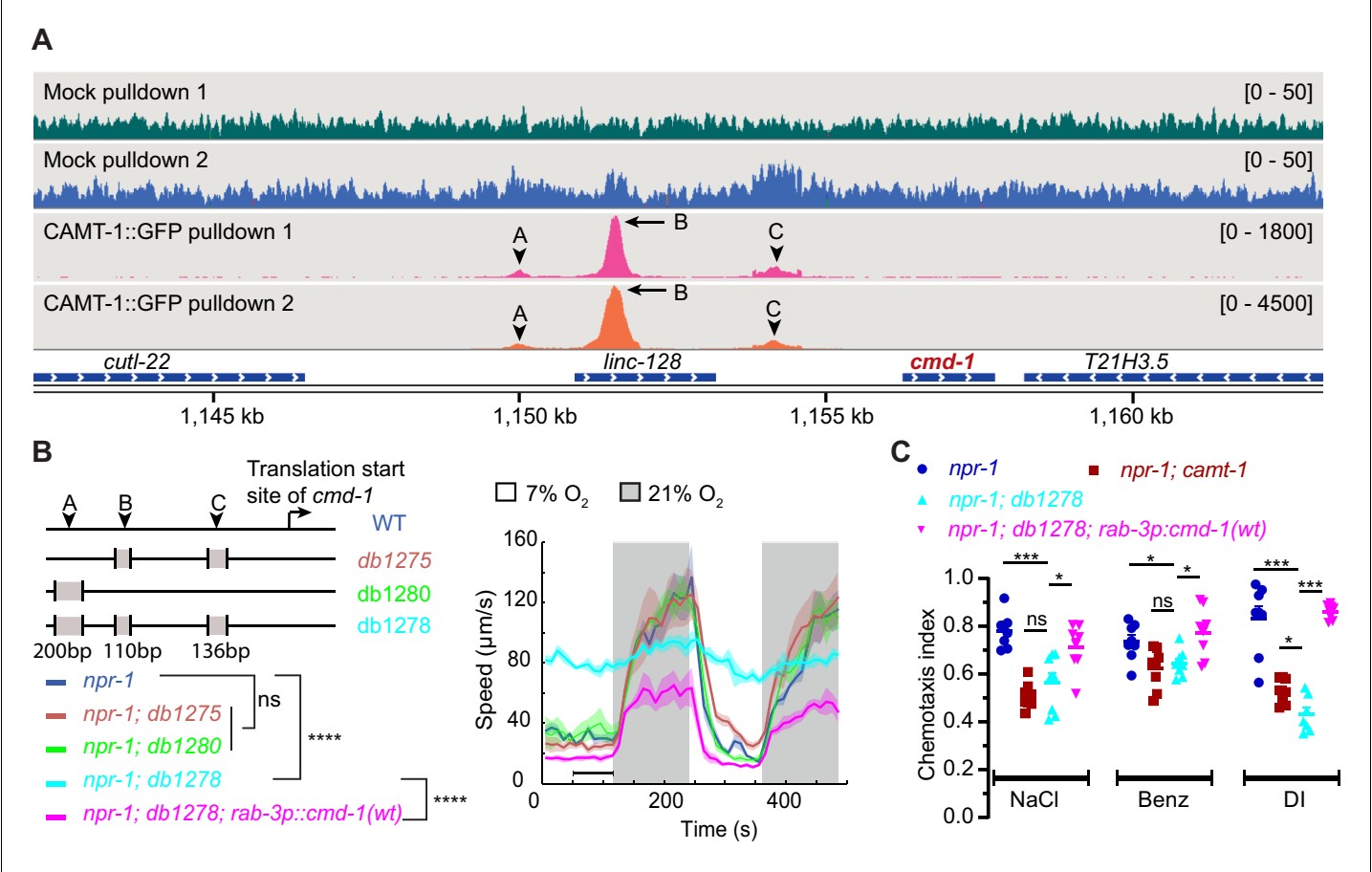

**Figure 6.** CAMT-1 directly activates calmodulin expression by binding multiple sites in the *cmd-1* promoter. (A) Coverage plots of chromatin pulldown samples showing enrichment at *cmd-1* promoter in CAMT-1::GFP pulldown (peaks A, B, and C; arrows: major peaks, arrow heads: minor peaks) compared to a mock pulldown or input (see also *Figure 6—figure supplement 1A*). Bracketed numbers on the right indicate the scale (normalized read counts). (B) Left: CRISPR-generated strains deleted for one or more of the CAMT-1 ChIP-seq peaks A, B, and C shown in (A); deletions are not drawn to scale. Right: O$_2$-evoked speed responses of the promoter deletion strains shown at left. The *db1278* allele in which all three CAMT-1 peaks are deleted confers a strong phenotype that can be rescued by supplementing CMD-1 expression in the nervous system. The *db1275* and *db1280* alleles, which delete only one or two sites have no obvious phenotype. (C) The *db1278* allele confers chemotaxis defects to NaCl, benzaldehyde, and diacetyl, similarly to *camt-1(ok515)* mutants, that can be rescued by supplementing CMD-1 expression in the nervous system. ns: p≥0.05, *: p<0.05, ***: p<0.001, ****: p<0.0001, Mann-Whitney U-test. n=2 (A), n≥49 (B), n=8 assays for each condition (C). (B) Lines represent average speed and shaded regions the SEM, black horizontal bars indicate time points used for statistical tests. (C) Colored bars indicate the mean and error bars indicate the SEM. ChIP-seq, chromatin immunoprecipitation sequencing.

The online version of this article includes the following figure supplement(s) for figure 6:

**Figure supplement 1.** Control samples for ChIP-seq and CAMT-1 binding peak in Y41C4A.17 promotor region.

and *rab-3p::cmd-1* showed neuronal GFP levels similar to those found in control animals lacking the *rab-3p::cmd-1* transgene (*Figure 7—figure supplement 1B*). These data suggest that CMD-1/CaM can negatively regulate its own expression by binding the IQ domains of CAMT-1. Thus, CAMT-1 may not only activate *cmd-1* expression, but also repress it when available CMD-1/CaM levels are high (*Figure 7*).

## Discussion

We find that neuronal levels of CaM, a key mediator of Ca$^{2+}$ signaling, are controlled by the CaM-binding transcriptional activator CAMTA in both *C. elegans* and *Drosophila*. Reduced CaM levels appear to explain the pleiotropic phenotypes of *C. elegans camt-1* mutants. First, *camt-1*

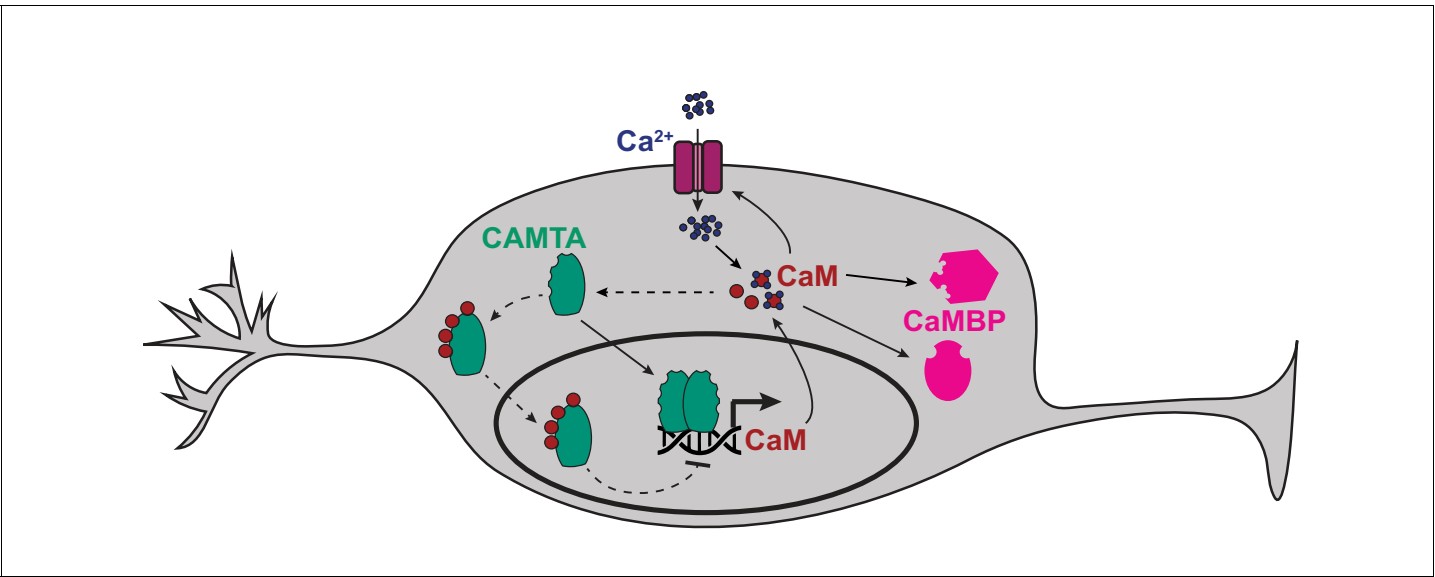

**Figure 7.** Model of how CAMT-1 may positively and negatively regulate levels of CaM in neurons. The binding of four apo-CaM to CAMTA is hypothetical, and is based on published data obtained from plant and *Drosophila* CAMTAs. CaMBP: Other CaM-binding proteins. Further analysis is required to confirm if the negative feedback loop occurs at physiological CaM concentrations. CaM, calmodulin; CAMTA, CaM-binding transcription activator.

The online version of this article includes the following figure supplement(s) for figure 7:

**Figure supplement 1.** CAMT-1 can repress CMD-1/CaM expression at high CMD-1/CaM levels.

phenotypes can be rescued by supplementing neurons with CaM. Second, deleting CAMT-1 binding sites in the CaM promotor phenocopies *camt-1*.

Profiling four different *C. elegans* neurons from *camt-1* mutants and WT controls using FAC sorting and RNA Seq shows that CAMT-1 stimulates CaM expression in each of the four neurons. These results, together with the observation that CAMT-1 is expressed in most or all *C. elegans* neurons, suggest that CAMT-1 is part of a general mechanism that regulates CaM levels throughout the nervous system.

The RNA Seq experiments reveal a 2.5×–4× reduction in CaM mRNA levels in *camt-1* mutants, depending on neuron type. These relatively small decreases in CaM mRNA are, however, associated with striking alterations in the stimulus-evoked $Ca^{2+}$ responses of each neuron. These findings suggest neural function is sensitive to quite small changes in CaM transcription. CaM levels may therefore provide a sensitive point of regulation of neural physiology. The increase in neuronal $Ca^{2+}$ levels we observe in the sensory neurons of *camt-1* mutants could simply reflect a decrease in $Ca^{2+}$ buffering by CaM. An alternative explanation for the $Ca^{2+}$ imaging phenotypes is that reducing CaM levels disrupts the regulation of $Ca^{2+}$/CaM's myriad binding partners. Previous work has identified multiple $Ca^{2+}$-CaM feedback loops regulating *C. elegans* sensory responses, mediated for example by calcineurin/TAX-6 (*Kuhara et al., 2002*), CaM kinase I/CMK-1 (*Satterlee et al., 2004*), and PDE1/PDE-1 (*Couto et al., 2013*). In addition, work in vertebrates (*Saimi and Kung, 2002*) has shown that CaM regulates the activity of cyclic nucleotide-gated ion channels and the L-type–$Ca^{2+}$ channel, which contribute to the $Ca^{2+}$ responses of these *C. elegans* sensory neurons. Further experiments are required to understand in mechanistic terms how altered CaM levels alter $Ca^{2+}$ signaling in *camt-1* mutants.

Profiling of $O_2$ sensors revealed that many genes showed altered expression in *camt-1* mutants compared to controls. Our analysis of mutants that abolish $O_2$-evoked $Ca^{2+}$ responses in these neurons shows that most of these expression changes are linked to increased $Ca^{2+}$ levels in *camt-1* mutants, rather than loss of CAMT-1 per se. This is consistent with the known role of $Ca^{2+}$ in regulating neuronal transcription (*Yap and Greenberg, 2018*). Our ChIP-seq studies identified CMD-1 as one of the major direct targets of CAMT-1. While binding motif analysis of the ChIP-seq data using

prediction tool MEME did not find hits that coincide with CAMT-1 binding sites at the *cmd-1* promoter(data not shown), we note that there are four mouse CAMTA1 binding motifs (*Long et al., 2014*; *Long et al., 2009*) overlapping with the CAMT-1 binding peaks of the *cmd-1* promoter.

CAMTA regulates CaM expression not only in *C. elegans* but also in *Drosophila*. Mutations in the sole *Drosophila* CAMTA, *dCAMTA*, cause an approximately two fold reduction in CaM mRNA and protein in the *Drosophila* head. These results suggest that the regulation of CaM expression by CAMTA proteins is conserved across phylogeny. Conservation may extend beyond metazoa, as in *Arabidopsis*, CAMTA3/AtSR1 binds in vitro to the promoter of CaM2, although whether this regulates CaM2 expression in vivo is unknown (*Yang and Poovaiah, 2002*).

Like CAMT-1, dCAMTA is expressed broadly in the nervous system (*Sato et al., 2019*). Previous work found that dCAMTA mutants have defective termination of photoresponses (*Han et al., 2006*). A separate study showed that a promoter mutation in the fly CaM gene that reduces CaM expression also disrupts photoresponse termination in *Drosophila* photoreceptors (*Scott et al., 1997*). Since *dCAMTA* mutants show reduced levels of CaM in photoreceptors (although not to the same extent as the promoter mutation), part of the photoresponse termination defect in these animals may reflect reduced levels of CaM. More generally, it would be interesting to ask if supplementing neuronal CaM levels can rescue the *dCAMTA* behavioral phenotypes.

Mammals encode two CAMTA genes, *CAMTA1* and *CAMTA2*. *CAMTA1* is expressed broadly in both the mouse and human nervous systems. Homozygous mice and heterozygous human patients bearing mutations in *CAMTA1* exhibit pleiotropic behavioral phenotypes, including memory defects and neurodegeneration (*Han et al., 2006*; *Sato et al., 2019*; *Long et al., 2014*; *Bas-Orth et al., 2016*; *Thevenon et al., 2012*; *Huentelman et al., 2007*). Our work raises the possibility that these defects are functionally associated with a reduction in CaM expression (*Zalcman et al., 2018*; *Wayman et al., 2008*). CAMTA2 is expressed in cardiomyocytes, and is implicated in promoting cardiac growth: overexpressing CAMTA2 in the mouse heart leads to cardiac hypertrophy (*Song et al., 2006*). Selectively overexpressing CaM in the mouse heart also induces cardiac hypertrophy, by a calcineurin-dependent mechanism (*Obata et al., 2005*). It would be interesting to ask if the cardiac hypertrophy in CAMTA2 overexpressing mice reflects increased CaM levels.

While CAMTAs were initially characterized as transcriptional activators, they have also been shown to mediate transcriptional repression (*Du et al., 2009*; *Kim et al., 2017*; *Sun et al., 2020*). Our data suggest CAMT-1 not only promotes CaM expression in *C. elegans* neurons, but can also inhibit it when available CaM levels are high, by a feedback loop in which CaM regulates its own transcription by binding to IQ domains of CAMT-1. These data suggest CAMT-1 can play a homeostatic role in regulating CaM levels (*Figure 7*). Mutant analyses in plants and flies have already suggested that CaM binding regulates CAMTA activity (*Du et al., 2009*; *Nie et al., 2012*; *Kim et al., 2017*; *Choi et al., 2005*). Our data suggest that binding to CaM converts CAMT-1 from an activator to a repressor. However, more data are required to establish if this feedback is relevant under physiological conditions. The absence of an obvious behavioral phenotype in mutant animals in which CAMT-1's four IQ motifs have been disrupted suggests that native CaM levels may simply not be high enough in the circuits we have studied to evoke negative feedback regulation of CaM expression.

In summary, our data suggest that we have discovered a general and conserved mechanism by which neurons control levels of CaM using CAMTA, a transcription factor that is expressed broadly in the nervous system across Metazoa. Toggling CAMT-1, the *C. elegans* CAMTA, up and down, can change neural excitability, circuit function, and behavior. We speculate that the activity of CAMTA transcription factors is regulated in response to upstream signals, and provides a mechanism to alter CaM levels and thereby modulate neural excitability and behavior.

## Materials and methods

No statistical methods were used to predetermine sample sizes. The sample size and replicate number were similar to or greater than that used in previously published papers (behavior assays, $Ca^{2+}$ imaging) or in the scientific literature (RNA-seq, ChIP-seq, Western blot, and qPCR). The experiments were not randomized. This work used only biological replicates (biologically distinct samples that capture random biological variation) but not technical replicates (repeated measurements from the same sample).

## Strains

*C. elegans* strains used are listed in *Supplementary file 4*. Strains were maintained at room temperature (RT) (22°C), on nematode growth medium (NGM) with *E. coli* OP50 unless otherwise specified. RB746 *camt-1*(*ok515*) and OH10689 *otIs355[rab-3p::2xNLS::TagRFP]* were obtained from the *Caenorhabditis* Genetic Center (P40 OD010440).

## Molecular biology

We obtained a clone containing the *camt-1* locus from the *C. elegans* fosmid library (Source BioScience). To insert GFP immediately prior to the termination codon of *camt-1* we followed established protocols (*Tursun et al., 2009*). The primers used to amplify the recombineering cassette from pBALU1 were: ATCATCCATGGGACCAATTGAAACCGCCGTATGGTTGCGGAACACTTGCAATGAG TAAAGGAGAAGAACTTTTCAC and aaaccaataaaaaaaatcggcatcttctaaaagtgacaccggggcaaTTATTTG TATAGTTCATCCATGCCATG. To generate transgenic lines, we injected a mix of 50 ng/µl fosmid DNA and 50 ng/µl co-injection marker (unc-122p::dsRED).

*C. elegans* expression constructs were generated using MultiSite Gateway Recombination (Invitrogen) or FastCloning (*Li et al., 2011*). We amplified cDNA corresponding to *camt-1* (*T05C1.4b*) using primers ggggACAAGTTTGTACAAAAAAGCAGGCTtttcagaaaaATGAATAATTCAGTCACTCG TCTTCTTTTCAAACGACTGCTGAC and ggggACCACTTTGTACAAGAAAGCTGGGTATTATGCAAG TGTTCCGCAACCATACGGCG. We were unable to amplify *camt-1* cDNA corresponding to the longer *T05C1.4a* splice variant so we generated it by site-directed mutagenesis of *T05C1.4b* cDNA. To convert *T05C1.4b* cDNA to *T05C1.4a* we used the Q5 Site-Directed Mutagenesis Kit (NEB) and primers gtcatactcaacatctaATTGCGGAAAATGCATGC and catcatcaatatttacaTTATTACGATTTTG TCGCATAAAATTC.

## Genome editing

Strains PHX994 and PHX1919 were generated by SunyBiotech at our request (Fujian, China). We generated point mutations in the endogenous *camt-1* locus using published CRISPR protocols (*Dokshin et al., 2018*). Cas9 endonuclease, crRNA, and tracrRNA were obtained from IDT (Iowa).

## Behavioral assays

$O_2$- and $CO_2$-response assays were performed as described previously (*Flynn et al., 2020*), using young adults raised at RT. 15–30 young adults were assayed in a microfluidic PDMS chamber on an NGM plate seeded with 20–50 µl OP50. The indicated $O_2$/$CO_2$ mixtures (in nitrogen) were bubbled through $H_2O$ and pumped into the PDMS chamber using a PHD 2000 Infusion Syringe Pump (Harvard Apparatus). Videos were recorded at two fps using FlyCapture software (FLIR Systems), and a Point Gray Grasshopper camera mounted on a Leica MZ6 microscope. Custom MATLAB software (Zentracker: https://github.com/wormtracker/zentracker, *Laurent et al., 2015*) was used to measure speed and omega turns.

Chemotaxis assays were performed as previously described (*Bargmann et al., 1993*) with minor modifications. 9 cm assay plates were made with 2% Bacto Agar, 1 mM $CaCl_2$, 1 mM $MgSO_4$, and 25 mM $K_2HPO_4$ pH 6. Test and control circles of 3 cm diameter were marked on opposite sides of the assay plate, equidistant from a starting point where >50 animals were placed to begin the assay. For olfactory assays, 1 µl odorant (Benzaldehyde 1/400 or Diacetyl 1/1000 dilution in ethanol) or 1 µl ethanol, and 1 µl 1M $NaN_3$, were added to each circle. For gustatory assays, an agar plug containing 100 mM NaCl was added the night before to the assay plates and removed prior to assay. Assays were allowed to proceed for 30–60 min, after which point plates were moved to 4°C, to be counted later. The chemotaxis index was calculated as (number of animals in test circle−number of animals in control circle)/total number of animals that have left the starting area.

## Heat-shock

Animals were raised at 20°C to reduce leaky expression from the *hsp-16.41* heat-shock promoter. To induce heat-shock, parafilm-wrapped plates were submerged in a 34°C water bath for 30 min, and then recovered at 20°C for 10 hr.

## Ca²⁺ imaging

Neural imaging was performed as previously described (*Flynn et al., 2020*), with a 2× AZ-Plan Fluor objective (Nikon) on a Nikon AZ100 microscope fitted with ORCA-Flash4.0 digital cameras (Hamamatsu). Excitation light was provided from an Intensilight C-HGFI (Nikon), through a 438/24 nm filter and an FF458DiO2 dichroic (Semrock). Emission light was split using a TwinCam dual-camera adapter (Cairn Research) bearing a filter cube containing a DC/T510LPXRXTUf2 dichroic and CFP (483/32 nm) and YFP (542/27) filters. We acquired movies using NIS-Elements (Nikon), with 100 ms or 500 ms exposure time. YFP/CFP ratios in URX were reported by YC2.60 driven from the *gcy-37* promoter, in BAG by YC3.60 and TN-XL driven from the *flp-17* promoter, in AFD by YC3.60 driven from the *gcy-8* promoter.

## Single-neuron-type cell sorting and RNA sequencing

We used *C. elegans* lines in which neuronal types were labelled by expressing GFP under specific promoters: oxygen sensing neurons (*gcy-37p*), BAG (*flp-17p*), RMG (combination of *ncs-1p::CRE* and *flp-21::loxP::STOP::loxP::GFP*; *Macosko et al., 2009*), and AFD (*gcy-8p*). These markers were crossed into either *npr-1(ad609)* or *npr-1(ad609); camt-1(ok515)* backgrounds. *C. elegans* cells were dissociated and GFP-labelled neurons were sorted as described previously (*Kaletsky et al., 2018*). Briefly, *C. elegans* with GFP-labelled neurons were synchronized using the standard bleaching protocol 3 days before the cell sorting and the eggs were placed on 90 mm rich NGM plates (7.5 g peptone/liter) seeded with OP50. For each sample, we used >50,000 worms. Worms were washed three times with M9, prewashed, and then incubated for 6.5 min with 750 µl lysis buffer (0.25% SDS, 200 mM DTT, 20 mM HEPES pH 8.0, and 3% sucrose). The worms were then rapidly washed five times with M9. We dissociated the cells by adding 500 µl of Pronase (Roche) 20 mg/ml and by either pipetting up-and-down or stirring continuously for 12 min using a small magnetic stirrer. The pronase was inactivated by adding 500 µl of phosphate-buffered saline (PBS)+2% fetal bovine serum (FBO) (Gibco). The solutions were passed through a 5 µm pore size syringe filter (Millipore), and filtered cells were further diluted in PBS+2% FBS for sorting using a Sony Biotechnology Synergy High Speed Cell Sorter. Gates for detection were determined using cells prepared in parallel from non-fluorescent animals using the same protocol. An average of 3000 cells was collected for each library, and sorted directly into lysis buffer containing RNAse inhibitor (NEB E6420). cDNA libraries were made from RNA using NEB's Next Single Cell/Low Input RNA Library Prep Kit for Illumina (NEB E6420). Libraries were sequenced on an Illumina HiSeq 4000 with single-end reads of 50 bases.

## Confocal microscopy and image analysis

Young adult worms were mounted for microscopy on a 2% agar pad in 1 M sodium azide. Image analysis and fluorescence quantification were carried out using Fiji (ImageJ, Wayne Rasband, NIH). The expression pattern of CAMT-1(fosmid)-GFP was imaged as previously described (*Flynn et al., 2020*) on an Inverted Leica SP8 confocal microscope using a 63×/1.20 N.A. water-immersion objective. Lines expressing a *cmd-1* transcriptional reporter (*cmd-1p::gfp*) and a red neuronal marker (either *rab-3p::mCherry* or *rab-3p::cmd-1::SL2::mCherry*) were imaged on an LSM800 inverted microscope (Zeiss) using a 63x/1.40 N.A. oil-immersion objective. The region between the two pharyngeal bulbs (*Figure 7—figure supplement 1A*) was imaged using stacks with a step size of 0.3 µm. A 3 µm section (10 images) around the middle of the pharynx was projected using the maximum projection method. Neurons were identified by thresholding the intensity of the red marker (mCherry). The neuronal regions overlapping with the pharynx or body wall muscles were excluded. The relative fluorescence in (*Figure 7—figure supplement 1B*) was defined as the GFP level in neurons minus background fluorescence divided by the level of fluorescence in the pharynx (metacorpus+isthmus +terminal bulb) minus background fluorescence.

Images of fly retinae were acquired using a Zeiss LSM800 microscope with a 20× objective. Only retinae oriented so that the long axis of the rhabdomeres was visible were selected for quantitative analysis. A representative region of the image, as shown in *Figure 5D1' and D2'*, was thresholded to segment the rhabdomeres, and the mean fluorescence intensity was measured, corrected to the background fluorescence, and plotted.

## Chromatin immunoprecipitation sequencing

The ChIP-seq protocol used is described in Wormbook (http://www.wormbook.org/chapters/www_chromatinanalysis/chromatinanalysis.html). Briefly, mixed-stage worms were grown in liquid culture, harvested, washed three times in PBS, and resuspended in PBS+Protease Inhibitor (PI, Sigma-Aldrich). Worm 'popcorn' was prepared by dripping worm solution into liquid nitrogen, and then hand ground to a fine powder. For each ChIP replicate we used 2.5 g of packed worms. Crosslinking was carried out by incubating samples in 1.5 mM EGS in PBS for 10 min, then adding 1.1% formaldehyde and incubating for a further 10 min. The reaction was quenched using 0.125 M glycine. The pellet was washed once in PBS+PMSF 1 mM and once in FA buffer (50 mM HEPES/KOH pH 7.5, 1 mM EDTA, 1% Triton X-100, 0.1% sodium deoxycholate, and 150 mM NaCl)+PI. The pellet was resuspended in 4 ml of FA buffer+PI+0.1% sarkosyl and sonicated using a Diagenode Bioruptor Plus with 40 cycles, 30 s on, 30 s off. The sample was then spun in a tabletop microcentrifuge at top speed (15,000 rpm) for 15 min. The supernatant was incubated with 1 µl of anti-GFP antibody from Abcam (Abcam Cat# ab290, RRID:AB_303395) overnight at 4°C. 60 µl of Protein A conjugated Dynabeads was added and the resulting solution incubated for 3 hr at 4°C. Pulldown, washing, and de-crosslinking steps were as described in http://www.wormbook.org/chapters/www_chromatinanalysis/chromatinanalysis.html. For preparing ChIP libraries, we used NEBNext Ultra II DNA Library Prep Kit for Illumina with half of the pulldown and 30 ng of input. DNA libraries were then sequenced on an Illumina HiSeq 4000 platform with single read of 50 bases.

## RNA-seq and ChIP-seq data analyses

RNA-seq data were mapped using PRAGUI—a Python 3-based pipeline for RNA-seq data analysis available at https://github.com/lmb-seq/PRAGUI (RRID:SCR_021692) . PRAGUI integrates RNA-seq processing packages including Trim Galore, FastQC, STAR, DESeq2, HTSeq, Cufflinks, and MultiQC. Output from PRAGUI was analyzed using PEAT—Pragui Exploratory Analysis Tool (https://github.com/lmb-seq/PEAT; RRID:SCR_021691) to obtain the list of differentially expressed genes with a false discovery rate<0.05. The Venn diagram was drawn using the online tool http://bioinformatics.psb.ugent.be/webtools/Venn/.

ChIP-seq data were analyzed using a nucleome processing and analysis toolkit that contains an automated ChIP-seq processing pipeline using Bowtie2 mapping and MACS2 peak calling. The software is available on Github at https://github.com/tjs23/nuc_tools (*Stevens, 2021*). Comparisons between different ChIP-seq conditions were carried out using the DiffBind package (*Stark and Brown, 2011*). ChIP-seq processed data were visualized using IGV (*Robinson et al., 2011*; *Thorvaldsdóttir et al., 2013*).

## Fly genetics

($w^{1188}$), ($w^{1118}$; $cn^1$, $tes^2$/cyo), and ($w^{1118}$; cro/cyo; sb/TM3 ser) flies were generously obtained from Daria Siekhaus (IST Austria), Hong-Sheng Li (UMass), and Daisuke Yamamoto (NICT), respectively. $cn^1$ flies were obtained from the Bloomington *Drosophila* Stock Center (NIH P40OD018537). These flies were crossed to obtain $w^{1118}$; $cn^1$ and $w^{1118}$; sb control flies.

## Quantitative PCR

qPCR was performed using the Janus Liquid Handler (PerkinElmer) and a LightCycler 480 system (Roche). Total RNA was extracted from the heads of 20 male adults or 17 female adults using a Monarch Total RNA Miniprep Kit (NEB). Three replicates for male and three replicates for female flies were done for each genotype. RNA was reverse transcribed into cDNA using an ImProm-II Reverse Transcription System (Promega). cDNA was mixed with Luna Universal qPCR Master Mix (NEB). *RpL32* (*rp49*) was amplified as an internal control. Primer sequences for *Rpl32* and *CAMTA* were identical to those used in *Sato et al., 2019*. *CaM* was amplified using the primer pair 5′-TGCAG-GACATGATCAACGAG-3′ (forward) and 5′-ATCGGTGTCCTTCATTTTGC-3′ (reverse). Data processing was performed using LightCycler Software (Roche).

## Western blot

Protein from the heads of ~50 female and 60 male adult flies were extracted using RIPA buffer (150 mM NaCl, 1% NP40, 0.5% sodium deoxycholate, 0.1% SDS, 50 mM Tris-HCl, pH 8.0, and PIs). Three

replicates for male and three replicates for female flies were performed for each genotype. After SDS-PAGE using Bolt 4–12% Bis-Tris Plus gels (Thermo Fisher Scientific), protein was transferred to PVDF membrane (0.45-μm pore size, Thermo Fisher Scientific) using the TE 22 Mighty Small Transfer Unit (Amersham Biosciences). Membranes were blocked with casein blocking buffer (1% Hammersten casein, 20 mM Tris-HCl, and 137 mM NaCl) for 1 hr, then incubated with primary antibody overnight at 4℃, followed by secondary antibody for 1 hr at RT. Unbound antibody was washed away with TBS-T or TBS (3× for 5 min). α-tubulin was used as an internal control. The following commercially available antibodies were used: anti-CaM (Abcam Cat# ab45689, RRID:AB_725815, diluted 1/500), anti-α-tubulin (Abcam Cat# ab40742, RRID:AB_880625, diluted 1/5000), goat anti-rabbit StarBright Blue 700 (Bio-Rad Cat# 12004161, RRID:AB_2721073, diluted 1/5000), and goat anti-mouse StarBright Blue 520 (Bio-Rad, 12005867, diluted 1/5000). Blots were imaged using the Chemidoc MP Imaging System (Bio-Rad).

### Immunostaining

Isolated retinae were dissected into ice-cold PBS, then fixed for 1 hr at 4℃ in 4% paraformaldehyde in PBS. Retinae were then rinsed in PBT (PBS, 0.5% Triton X-100) and incubated in the same solution for 3 days at 4℃ to wash out eye pigments, then blocked in PBT+10% Normal Goat Serum for 15–20 min. Retinae were subsequently incubated in primary antibodies mouse anti-CaM 1:200 (Invitrogen MA3-918, RRID:AB_325501) 1:200 at 4℃ for 3 days. After several washes in PBT, retinae were incubated with secondary antibodies (1:500 goat anti-mouse: Alexa Fluor 546, A-11030, RRID:AB_2534089) for 3 days at 4℃. Retinae were again washed three times for 15 min, with DAPI 1:1000 Thermo Fisher Scientific 62248 included in the second wash, mounted in Vectashield.

### Statistical tests

Statistical tests were two-tailed and were performed using Matlab (MathWorks, MA), GraphPad Prism (GraphPad Software, CA, RRID:SCR_002798), or R (R Foundation for Statistical Computing, Vienna, Austria, RRID:SCR_001905, http://www.R-project.org/). Measurements were done from distinct samples.

## Acknowledgements

The authors thank the MRC-LMB Flow Cytometry facility and Imaging Service for support, the Cancer Research UK Cambridge Institute Genomics Core for Next Generation Sequencing, Julie Ahringer and Alex Appert for advice and technical help for ChIP-seq experiments, Paula Freire-Pritchett, Tim Stevens, and Gurpreet Ghattaoraya for RNA-seq and ChIP-seq analyses, Nikos Chronis for the TN-XL plasmid, Hong-Sheng Li and Daisuke Yamamoto for generously sending the $tes^2$ and $cro$ mutants, Daria Siekhaus for hosting the fly work, Michaela Misova for technical assistance. This work was supported by an Advanced ERC grant (269058 ACMO) and a Wellcome Investigator Award (209504/Z/17/Z) to MdB, and an IST Plus Fellowship to TV-B (Marie Sklodowska-Curie Agreement no 754411).

## Additional information

### Funding

| Funder | Grant reference number | Author |
| --- | --- | --- |
| Wellcome | 209504/A/17/Z | Mario de Bono |
| European Research Council | 269058 ACMO | Mario de Bono |
| H2020 Marie Skłodowska-Curie Actions | 754411 | Thanh Thi Vuong-Brender |
| Medical Research Council | Studentship | Sean Flynn |

The funders had no role in study design, data collection and interpretation, or the decision to submit the work for publication.

## Author contributions
Thanh Thi Vuong-Brender, Conceptualization, Formal analysis, Funding acquisition, Investigation, Writing - original draft, Writing - review and editing; Sean Flynn, Conceptualization, Formal analysis, Investigation, Writing - original draft, Writing - review and editing; Yvonne Vallis, Formal analysis, Investigation, Methodology, Yvonne Vallis contributed to the revision experiments of this manuscript; Saliha E Sönmez, Investigation;  Mario de Bono, Conceptualization, Supervision, Funding acquisition, Visualization, Writing - original draft, Project administration, Writing - review and editing

## Author ORCIDs
Thanh Thi Vuong-Brender (ID) https://orcid.org/0000-0001-6594-2881
Sean Flynn (ID) http://orcid.org/0000-0001-7326-2659
Mario de Bono (ID) https://orcid.org/0000-0001-8347-0443

## Decision letter and Author response
Decision letter https://doi.org/10.7554/eLife.68238.sa1
Author response https://doi.org/10.7554/eLife.68238.sa2

## Additional files

### Supplementary files
• Supplementary file 1. The 100 most highly expressed genes (in order of decreasing read counts, in TPM) from neuron-specific RNA profiling.

• Supplementary file 2. Genes showing differential expression between *camt-1* and WT in the profiled neural types.

• Supplementary file 3. Genomic locations differentially bound by CAMT-1 identified using the Diff-Bind algorithm for ChIP-seq data with a false discovery rate (FDR) threshold of 0.05. Genes overlapping or within 10 kb downstream of these sites are reported. The table is sorted in the order of increasing FDR. Note that the CAMT-1 binding site at the *cmd-1* promoter was annotated with the overlapping long intervening noncoding RNA *linc-128*.

• Supplementary file 4. List of *C. elegans* strains used in this study.

• Supplementary file 5. Exact p-values and n numbers for experiments reported in this study.

• Transparent reporting form

### Data availability
Sequencing data have been deposited in GEO under accession codes GSE164671.

The following dataset was generated:

| Author(s) | Year | Dataset title | Dataset URL | Database and Identifier |
|---|---|---|---|---|
| Vuong-Brender TT, Flynn S, Bono M | 2020 | Transcriptional control of CALMODULIN by CAMTA regulates neural excitablity | https://www.ncbi.nlm.nih.gov/geo/query/acc.cgi?acc=GSE164671 | NCBI Gene Expression Omnibus, GSE164671 |

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
