## [Decision Letter]

**Acceptance summary:**

Calcium-calmodulin (CaM) signaling plays an essential role within and outside of the nervous system. Moreover, it is conserved from plants to humans. While a lot is known about the mechanisms of cellular calcium level fluctuations, how CaM levels are regulated is less clear. In this manuscript, Vuong-Brender and colleagues characterize a, likely, conserved role of the transcription factor CAMT-1 in the homeostatic regulation of CaM levels and show how it impacts animal behavior and nervous system function. The paper is a tour-de-force across multiple techniques and model systems. The data is of a very high quality and supports most of the authors’ claims strongly and convincingly.

**Decision letter after peer review:**

Thank you for submitting your article "Camta tunes neural excitability and behavior by modulating calmodulin expression" for consideration by *eLife*. Your article has been reviewed by 3 peer reviewers, one of whom is a member of our Board of Reviewing Editors, and the evaluation has been overseen by Piali Sengupta as the Senior Editor. The reviewers have opted to remain anonymous.

The overall opinion is that your manuscript is interesting and potentially appropriate for the readership of *eLife*. Nevertheless, there were a few concerns that need to be addressed before a formal decision can be reached. The reviewers have discussed their reviews with one another, and the Reviewing Editor has drafted this to help you prepare a revised submission.

Essential revisions:

1) All three reviewers expressed concern regarding the model in Figure 7. In particular, the mechanism and role of the inhibitory function of Camt-1 require elucidation. We would like you to consider (a) providing more data strengthening the model (e.g. analyzing behavior of the syb1919 mutant in WT vs. npr1 vs. double transgenic background), (b) reducing the emphasis by moving the repressor function into supplementary data or (c) by removing it altogether. I should point out though that we would prefer option (a) or (b).

2) A second concern relates to the use of GECI imaging as they contain CaM domains. The reviewers suggest that this could be addressed by for instance using your reporter assay or behavioral analysis upon expression of the indicator.

3) Given that you are reporting a highly conserved phenomenon, we would like to encourage you to consider making some changes to the narrative to make the manuscript more accessible for a broader readership who might not be very familiar with methods and prior findings of the *C. elegans* field.

*Reviewer #1 (Recommendations for the authors):*

1. Your data is very convincing. Congratulations! I was, however, somewhat disappointed to not learn when and what might regulate CAMTA given that CaM levels appear to matter so much. Have you considered to analyze cmd-1 reporter expression at different internal or behavioral states? Given that CaM levels finetune neuronal sensitivity, would it be reasonable to assume that oxygen experience, aggregation state etc. influence reporter expression? Adding such data would make this manuscript, in my opinion, even more interesting.

2. You write, for instance: "despite CAMT-1 loss increasing Ca^2+^ levels in URX, Ca^2+^ responses in RMG were strongly reduced in camt-1 mutants". This is unexpected, and I wonder if the bidirectional regulation through CAMTA might explain it?

3. Similarly, it is still unclear how exactly CAMTA regulates calcium levels. Is the model that less CaM leads to more free calcium and more CaM to less free calcium?

4. Ca^2+^-CaM interacts with multiple pathways including binding to Camt-1. Can you please clarify whether Camt-1 is bound by Ca^2+^-bound or free CaM. Your scheme in figure 7 suggests the latter. However, I don't think the presented data provides evidence for this.

5. It is somewhat surprising Camt-1 IQ domain mutants show no behavioral phenotype suggesting that higher CaM levels do not matter. Could you please explain this a little better given that you say elsewhere that CaM levels are critical.

6. The description of figure 3 states: "camt-1 mutants show heightened stimulus-evoked Ca^2+^ responses in sensory neurons.". By contrast, and in my opinion reflecting the actual data, you describe your findings as increased basal calcium levels in camt-1 mutants. I suggest changing the description of figure 3.

7. Given that you are reporting a likely highly conserved mechanism, a somewhat less *C. elegans*-specific setting would make the paper more accessible to readers outside the field. While the text is very well written and fairly easy to understand, the figures are more difficult to comprehend due to the use of different genetic backgrounds (N2, npr-1) and the many different stimuli or stimulus concentrations. A reader from the plant or the mouse field will likely not be able to appreciate the meaning of, for instance, the different neurons you are recording from. Is there any way this could be simplified? Or better illustrated with a little model in the figures or in a figure?

*Reviewer #2 (Recommendations for the authors):*

The unresolved questions of why RMG calcium is affected by CAMT-1 mutation differently than sensory neuron calcium and how CAMT-1 acts as both a repressor and activator of calmodulin expression are given more prominence in this manuscript than warranted. The key discovery that effects of CAMT-1 mutation can be corrected by restoring expression of one key target, calmodulin, is not affected by these observations.

---

## [Author Response]

Essential revisions:1) All three reviewers expressed concern regarding the model in Figure 7. In particular, the mechanism and role of the inhibitory function of Camt-1 require elucidation. We would like you to consider (a) providing more data strengthening the model (e.g. analyzing behavior of the syb1919 mutant in WT vs. npr1 vs. double transgenic background), (b) reducing the emphasis by moving the repressor function into supplementary data or (c) by removing it altogether. I should point out though that we would prefer option (a) or (b).

We agree that the inhibitory function of CAMT-1 needs further experimental analysis. We have therefore reduced the emphasis on it, by moving the relevant data to a supplementary figure, and by explicitly saying that further experiments are needed to establish if this feedback is relevant under physiological conditions.

2) A second concern relates to the use of GECI imaging as they contain CaM domains. The reviewers suggest that this could be addressed by for instance using your reporter assay or behavioral analysis upon expression of the indicator.

We agree that this is an important point, especially given the ubiquitous use of Ca^2+^ sensors bearing CaM domains. We have sought to address it in multiple ways.

First, we decided to repeat some of our Ca^2+^ imaging experiments using a genetically-encoded Ca^2+^ indicator that does not contain CaM. We opted to use TN-XL, an indicator that uses troponin C as the Ca^2+^ binding moiety, and which has previously been used successfully in *C. elegans(1)*. We imaged CO_2_-evoked Ca^2+^ responses in BAG sensory neurons both in wild type and in *camt-1* mutant animals. The data obtained using TN-XL recapitulated what we observed using YC3.60 (BAG). The data has been incorporated in Figure 3 —figure supplement 1. We also tried to image O_2_-evoked Ca^2+^ responses in RMG interneurons using TN-XL but could not detect responses either in wild-type or mutant. This likely reflects the smaller size of O_2_-evoked Ca^2+^ responses in RMG, which we normally image with the high Ca^2+^ affinity sensor YC2.60. YC2.60 has higher Ca^2+^ sensitivity than TN-XL.

Second, we compared the behavioral responses of transgenic lines expressing GECIs that include CaM to non-transgenic controls. For the lines we used that express GECIs in BAG, AFD and URX we did not see any differences in the responses. These data are in Figure 3 —figure supplement 1. For the line expressing YC2.60 in RMG we did observe a difference – the YC2.60 transgene dampened the behavioral response. We have however removed the RMG Ca^2+^ imaging data, In the interest of simplifying the manuscript as advised by our reviewers, and so do not include the behavioral data.

3) Given that you are reporting a highly conserved phenomenon, we would like to encourage you to consider making some changes to the narrative to make the manuscript more accessible for a broader readership who might not be very familiar with methods and prior findings of the C. elegans field.

We thank our reviewers for this suggestion. We have asked 4 non-specialists to read through the manuscript and identify where we are cryptic and using *C. elegans*-centric phrasing. We have re-written the areas of text highlighted by our reviewers and our sounding boards, and hope we have succeeded in making the paper more accessible to a broad audience. It is difficult to avoid the genetic nomenclature.

Reviewer #1 (Recommendations for the authors):1. Your data is very convincing. Congratulations! I was, however, somewhat disappointed to not learn when and what might regulate CAMTA given that CaM levels appear to matter so much. Have you considered to analyze cmd-1 reporter expression at different internal or behavioral states? Given that CaM levels finetune neuronal sensitivity, would it be reasonable to assume that oxygen experience, aggregation state etc. influence reporter expression? Adding such data would make this manuscript, in my opinion, even more interesting.

The reviewer makes very good points: we are actually very eager to discover how and when CAMTA function is regulated. These are key questions we are currently exploring.

2. You write, for instance: "despite CAMT-1 loss increasing Ca^2+^ levels in URX, Ca^2+^ responses in RMG were strongly reduced in camt-1 mutants". This is unexpected, and I wonder if the bidirectional regulation through CAMTA might explain it?

This is a reasonable hypothesis to make, but we have not yet found evidence for it. Our current, speculative explanation for the RMG Ca^2+^ phenotype is that since communication between neurons can itself be upregulated by Ca^2+^/CaM (e.g. see Journal of Neuroscience 2010, 30, 4132-4142), one can imagine that lower CaM might reduce communication between URX and RMG. An alternative plausible hypothesis, is that the set of CaM binding proteins, for example ion channels, in URX and RMG are different, such that, reduced CaM in *camt-1* mutants has different effects on Ca^2+^ currents. In the interests of simplifying the manuscript, given the overall comments by our reviewers, we have removed the RMG Ca^2+^ imaging data.

3. Similarly, it is still unclear how exactly CAMTA regulates calcium levels. Is the model that less CaM leads to more free calcium and more CaM to less free calcium?

The model proposed by the reviewer, that changing CaM levels alters Ca^2+^ buffering, is one possibility, and we now say this explicitly in the discussion. An alternative explanation is suggested by data in the literature showing that CaM is limiting compared to the amounts of Ca^2+^/CaM binding proteins. We speculate that the substantial physiological effects of reducing CaM levels may principally reflect disrupted regulation of Ca^2+^/CaM’s binding partners. Many of these partners mediate negative feedback mechanisms that dampen or terminate Ca^2+^ signalling.

For example, we speculate that lower CaM availability will reduce activation of CaM kinases and calcineurin by Ca^2+^/CaM upon neuronal activation. Ca^2+^/CaM also binds and negatively regulates many ion channels, including cGMP channels and L-type Ca^2+^ channels. Lower levels of CaM (and therefore of Ca^2+^/CaM) will likely result in less effective feedback regulation of these channels.

We outline these ideas more explicitly in the discussion of the revised manuscript. We add that since we have not directly investigated the mechanisms by which changes in CaM levels leads to altered neural properties we are only speculating. We agree however that this is an important question. We also reference previous work that has implicated Ca^2+^/CaM binding targets like CaM Kinases and Calcineurin in regulating the properties of *C. elegans* sensory neurons.

4. Ca^2+^-CaM interacts with multiple pathways including binding to Camt-1. Can you please clarify whether Camt-1 is bound by Ca^2+^-bound or free CaM. Your scheme in figure 7 suggests the latter. However, I don't think the presented data provides evidence for this.

This is a good and interesting point. We have cited published data that give some indication of the complexity of CAMTA proteins as CaM and Ca^2+^/CaM binding proteins. In plant and *Drosophila* CAMTA, the IQ domains bind Ca^2+^-free CaM (*2, 3*), but they also have an additional domain that binds Ca^2+^/CaM(*2, 4, 5*). With regards to CAMT-1 itself, we need to address this question with biochemistry, i.e. by expressing and purifying the different predicted CaM-binding domains of CAMT-1 and studying their interaction with CaM +/- Ca^2+^. Performing such experiments would take some time, and since we have relegated regulation of CAMT-1 by CaM to Supplementary data, may be better suited to a study focussed on the regulatory significance of these binding properties. To make this point clear to readers, we added a note in the model in the figure 7 saying ‘Binding of four apo-CaM to CAMT-1 is hypothetical, based on data obtained from plant and *Drosophila* CAMTAs.’

5. It is somewhat surprising Camt-1 IQ domain mutants show no behavioral phenotype suggesting that higher CaM levels do not matter. Could you please explain this a little better given that you say elsewhere that CaM levels are critical.

This is a good point – we were also surprised by the behavioral observations. We interpret our results to indicate that native CaM levels may simply not be high enough in the circuits we have studied under our assay conditions to evoke negative feedback regulation

We have added a paragraph to the discussion to discuss this. We note in this section that in *Drosophila*, while disrupting the second and third IQ domains does not attenuate CAMTA’s ability to promote termination of light responses, disrupting the first IQ domain does.

6. The description of figure 3 states: "camt-1 mutants show heightened stimulus-evoked Ca^2+^ responses in sensory neurons.". By contrast, and in my opinion reflecting the actual data, you describe your findings as increased basal calcium levels in camt-1 mutants. I suggest changing the description of figure 3.

We have changed the description for Figure 3 to: ‘*camt-1* mutants show altered Ca^2+^ traces in sensory neurons’. It is clear that increasing CAMT-1 levels in URX (3C) and BAG (3D) reduces the Ca^2+^ baseline, whereas the Ca^2+^ baseline appears higher in *camt-1(null)* mutants. We think the stimulus-evoked responses also appear altered on several of the traces, but since the baseline is changed the relative contribution of the two is unclear.

7. Given that you are reporting a likely highly conserved mechanism, a somewhat less C. elegans-specific setting would make the paper more accessible to readers outside the field. While the text is very well written and fairly easy to understand, the figures are more difficult to comprehend due to the use of different genetic backgrounds (N2, npr-1) and the many different stimuli or stimulus concentrations. A reader from the plant or the mouse field will likely not be able to appreciate the meaning of, for instance, the different neurons you are recording from. Is there any way this could be simplified? Or better illustrated with a little model in the figures or in a figure?

We agree that *C. elegans* nomenclature can get in the way of the message. We hope our revised manuscript will help improve the manuscript’s clarity.

Reviewer #2 (Recommendations for the authors):The unresolved questions of why RMG calcium is affected by CAMT-1 mutation differently than sensory neuron calcium and how CAMT-1 acts as both a repressor and activator of calmodulin expression are given more prominence in this manuscript than warranted. The key discovery that effects of CAMT-1 mutation can be corrected by restoring expression of one key target, calmodulin, is not affected by these observations.

We have sought to put less emphasis on these points in the manuscript. As advised by our reviewer, we have removed the RMG Ca^2+^ imaging data that is not central to, and distracts from, our message.